# Characteristics, Components, and Efficacy of Telerehabilitation Approaches for People with Chronic Obstructive Pulmonary Disease: A Systematic Review and Meta-Analysis

**DOI:** 10.3390/ijerph192215165

**Published:** 2022-11-17

**Authors:** Sara Isernia, Chiara Pagliari, Luca Nicola Cesare Bianchi, Paolo Innocente Banfi, Federica Rossetto, Francesca Borgnis, Monica Tavanelli, Lorenzo Brambilla, Francesca Baglio

**Affiliations:** 1IRCCS Fondazione Don Carlo Gnocchi ONLUS, 20148 Milan, Italy; 2IRCCS Fondazione Don Carlo Gnocchi ONLUS, 50143 Firenze, Italy

**Keywords:** telerehabilitation, chronic obstructive pulmonary disease, continuity of care, digital medicine, pulmonary, rehabilitation

## Abstract

Introduction: Chronic obstructive pulmonary disease (COPD) is at the top of the list of non-communicable diseases with related rehabilitation needs. Digital medicine may provide continuative integrated intervention, overcoming accessibility and cost barriers. Methods: We systematically searched for randomized controlled trials on telerehabilitation (TR) in people with COPD to profile the adopted TR strategies, focusing on TR models and the main rehabilitation actions: monitoring and assessment, decision, and feedback. Additionally, a meta-analysis was run to test the TR effect on functional capacity, dyspnea, and quality of life compared to no intervention (NI) and conventional intervention (CI). Results: Out of the 6041 studies identified, 22 were eligible for the systematic review, and 14 were included in the meta-analyses. Results showed a heterogeneous scenario in terms of the TR features. Furthermore, only a small group of trials presented a comprehensive technological kit. The meta-analysis highlighted a significant effect of TR, especially with the asynchronous model, on all outcomes compared to NI. Moreover, a non-inferiority effect of TR on functional capacity and quality of life, and a superiority effect on dyspnea compared to CI were observed. Finally, the studies suggested a high rate of TR adherence and high safety level. Conclusions: TR is an effective strategy to increase and maintain functional capacity, breath, and quality of life in people with COPD. However, a consensus on the essential elements and features of this approach needs to be defined, and the effect of long-term maintenance merits further investigation.

## 1. Introduction

Chronic obstructive pulmonary disease (COPD) is a progressive respiratory illness characterized by airways and alveoli obstruction with chronic respiratory distress and related symptoms [1]. COPD is a leading cause of premature mortality and disability [1], at the top of the list of prevalent non-communicable diseases [2]. Prevalence is estimated between 4 and 10%, and mortality is about 3% [3]. Reducing the burden of COPD is one of the main aims of the Global Sustainable Development Goals [2]. For this purpose, pulmonary rehabilitation (PR) is currently an effective strategy in COPD management. A recent meta-analysis [4] on the effect on mortality of PR delivered as a supervised multidisciplinary program, including exercise training compared with usual post-exacerbation care or no PR program, showed reductions in mortality, the number of days in the hospital and the number of readmissions after PR in patients hospitalized with a COPD exacerbation.

Importantly, COPD needs an integrative and continuative approach to maintain autonomy in daily life [5]. The pulmonary rehabilitation statements recommend comprehensive interventions assuring rehabilitation, education, self-management, and psychological support [6,7,8]. However, due to healthcare costs, care system overburden, and distance barriers, a continuative and comprehensive approach is not always sustainable in the clinic, especially in the earliest phases of the disease. In addition, adherence to rehabilitation is a critical issue in this population [9] as a result of intrapersonal (e.g., lack of motivation), interpersonal (e.g., absence of a support system), and structural barriers (e.g., distance from the clinic, financial struggles) [10].

In response to this need, recent digital health models may constitute optimal solutions. Thanks to the adoption of mobile devices and healthcare platforms, the telerehabilitation (TR) approach allows care to overcome accessibility and cost barriers, potentially scaling up the service to a broader population [11,12,13]. A recent meta-analysis [14] on randomized and non-randomized trials supported the non-inferiority effect of TR compared to hospital-based interventions in the COPD population. Given these promising effects, the TR approach should be further investigated. In fact, despite the term “TR” having been previously referred to as any type of home-based therapy [15], the adoption of distinct models and different essential elements diversifies TR strategies.

Based on the communication modality, the TR model could be synchronous, asynchronous, or mixed. In the synchronous model, the communication between patients and clinicians is bidirectional and synchronized, and the interactions occur in real-time through videoconferencing systems, mimicking the *on-to-one* setting in the clinic. In asynchronous TR, the communication between patients and clinicians is temporally decoupled, allowing for bypassing of the 1:1 setting, which is typical of the *face-to-face* mode. Patients can perform rehabilitation activities without the therapist being online, who prescribes, plans, and monitors the rehabilitation activities by accessing the platform system. This model needs more complex digital infrastructure than synchronous interventions to ensure a good level of treatment personalization, making asynchronous TR comparable to hospital-based interventions [12]. In addition to the specific model, other features that distinguish different TR strategies are the decision and feedback type. The intervention’s adaptability over time requires the therapist to monitor and assess the patient’s TR activity to make decisions and set the dose and intensity of the sessions. Additionally, the feedback provided by the therapist to the patients to report the progression of the rehabilitation program can be online, during the course of the intervention, or offline, at the end of the program. Without these actions, the rehabilitation program is not personalized and adapted over time and may not be considered as comparable to a hospital-based treatment.

To the best of our knowledge, a systematic review that aims to characterize TR adopted for COPD, considering all rehabilitation approaches (TR model, assessment/monitoring, decision, feedback), and testing the effect of the asynchronous compared to synchronous approach is still lacking. 

This review aims to profile the strategies adopted for COPD TR and test their effectiveness on medical-benefit outcomes versus no intervention and/or an active comparator, such as conventional rehabilitation.

## 2. Methods

This systematic review and meta-analysis were realized according to the PRISMA (Preferred Reporting Items for Systematic Reviews and Meta-Analyses) guidelines [16]. 

### 2.1. Eligibility Criteria 

The inclusion criteria for studies’ eligibility were as follows: (1) primary studies; (2) testing the effect of TR; and (3) on people with COPD (≥18 years old). The exclusion criteria were as follows: (1) study design different from a randomized controlled trial (RCT); (2) non-English language; (3) pharmacological studies; (4) mixed sample (people with COPD and people with a different diagnosis; (5) no humans; (6) no planned rehabilitation activities; and (7) TR delivery out of the patient’s home.

### 2.2. Interventions

#### 2.2.1. Telerehabilitation

We refer to TR as home-based rehabilitation delivered through technology, assuring communication between the clinic and the patient at home. This type of communication may enable remote assessment and monitoring of the patient’s performance and response with appropriate feedback to the patient. In this approach, the presence of a ‘double loop’ communication is expected. In this case, the rehabilitation can be personalized and modified according to the actual progress/performance of the patient [17]. On the other hand, the lack of a ‘double loop’ renders the intervention equivalent to a prescription of home exercises without a real rehabilitative component that should include some interaction between patients and the medical team to adjust the patient’s exercise program and therapy. 

#### 2.2.2. Comparators

We collected data on two separate comparators: no intervention (NI), comprising no rehabilitation activities, and conventional intervention (CI), comprising planned exercise prescription with a similar dose of rehabilitation activities of TR. 

### 2.3. Information Sources for Study Selection

The data bench utilized for the literature search included MEDLINE, Scopus, and Web of Science. The search was performed on 26 September 2022, including all studies published from 2010 to September 2022 using the following keywords: ‘Tele*’ OR ‘virtual reality*’ OR ‘computerized treatment*’ OR ‘computerized training*’ OR ‘computer-assisted rehab*’ OR ‘serious game*’ OR ‘videogame*’ OR ‘home-based treatment*’ OR ‘home-based training*’ AND ‘chronic obstructive pulmonary disease’.

### 2.4. Selection and Data Collection Process

The eligibility screening was blindly and independently conducted by two researchers on the Rayyan platform [18]. Inter-reviewer discrepancies were solved by discussion to reach a consensus or by a third reviewer when the agreement was not reached (Figure 1). After the study selection, data collection was focused on the characteristics of the sample (demographics; inclusion and exclusion criteria), FITT descriptors (frequency, intensity, time, type), TR actions (monitoring/assessment, decision, and feedback), and type of technology (platform, device, digital content). In addition, data regarding outcome measures (patient-relevant and medical benefit) were also collected. 

### 2.5. Study Risk of Bias Assessment

To evaluate the RCT studies’ quality, the Tool for the assEssment of Study qualiTy and reporting in EXercise (TESTEX) scale [19] was blindly utilized by the two researchers to assess 12 criteria of external and internal validity of the trials’ design. For each TESTEX item, a score of 0–2 (criteria not satisfied-criteria addressed) was reported for a total score range of 0–15. A score of 12–15 suggests a study of high quality, a score of 7–11 good quality, and a score of ≤6 low quality.

### 2.6. Meta-Analysis and Narrative Synthesis

Statistical analyses were computed using RStudio software, version 3, adopting the metafor R package. Exercise capacity, dyspnea, and quality of life were considered fundamental outcome measures for COPD and included in the meta-analyses. First, the overall effect of TR on exercise capacity (6 m walk distance), dyspnea, and quality of life was computed. The unit of analysis considered was the standardized mean difference (SMD) of change from pre-treatment to post-treatment between TR and the control group. SMD as Hedges’*g* and 95% confidence intervals (95% CIs) were computed for outcomes of each study selected for the meta-analysis. The overall effect of TR on the specific outcome was pooled using a random-effect model. Corrections for inter-correlation among outcomes were assumed at 0 and 0.5. A *g* value ≤ 0.30, >0.30, and ≥0.60 was interpreted as a small, moderate, and high effect size according to Higgins et al. [20]. The proportion of true variance from the total observed variance was reported by I^2^ statistic with 95% CI (an I^2^ value of 25%, 50%, and 75% suggested a low, moderate, and high proportion of variance). Potential publication bias was investigated by reporting the funnel plot to detect eventual asymmetry and a small study effect, and the estimated number of missing studies was checked using the trim-and-fill procedure. Cochrane’s recommendations were strictly followed to overcome limitations related to missing values.

## 3. Results

### 3.1. Study Selection

Out of the 6041 studies identified, 22 studies were eligible for the systematic review. Among these, 14 works were considered for the meta-analysis, according to the availability of the raw data (Figure 1).

Based on the TESTEX results (Table 1), six trials were classified as high-, eleven as good-, and four as low-quality studies (mean TESTEX score = 9.76 ± 2.53). 

### 3.2. Participants

This review comprised 2234 COPD patients, 59% males, in a GOLD stage I-III. Among these, 1048 patients underwent TR, 683 CI, and 503 NI (Table 2). The inclusion and exclusion criteria of each study are reported in the Appendix A. All studies were conducted in developed countries. 

### 3.3. TR Interventions

Twenty trials provided an individual rehabilitation setting [8,21,22,23,24,25,26,27,28,29,30,31,32,33,34,35,36,37,38,39] while the rest of the trials [40,41] were in a group setting. Two trials [8,39] tested the maintenance effect of the TR intervention for the continuity of care after discharge from the clinic. Nine studies presented a unidimensional target (exercise training) [21,22,23,28,30,31,33,37,39] while thirteen studies a multi-dimensional one [8,24,25,26,27,29,32,34,35,36,38,40,41]. The treatment dose (FITT descriptors) was widely heterogeneous: (a) (T) the minimum duration was 6 weeks [26] versus a maximum of 52 weeks of treatment [8,24]; (b) (F) the frequency varied from 7 rehabilitation sessions per day [30] to 1 session per week [41]; (c) (T) time per session ranged from 10–20 [37] to 120 min [38]; and (d) (I) the intensity was set based on several modalities: Borg scale [23,27,29,33,40], step count [34], heart rate [22,25], Pi_max_ [24,28,30], or with a fixed incremental method [26] (Table 3). 

**Table 1 ijerph-19-15165-t001:** The Tool for the assEssment of Study qualiTy and reporting in EXercise (TESTEX) scale score for each study was included in the systematic review.

	1	2	3	4	5	6	7	8	9	10	11	12	Testex
	Eligibility	Randomization	Allocation	Groups Similarity at Baseline	Assessor Blinding	Outcome Measures	Intention-to-Treat	Between-Group Statistical Comparison	Point Measures and Measures of Variability	Activity Monitoring in the Control Group	Exercise Intensity Remained Constant	Exercise Volume and Energy Expenditure	
Ref. [36]	1	1	0	1	0	2	0	2	1	0	1	0	9
Ref. [26]	1	1	1	1	1	1	1	2	1	0	1	1	12
Ref. [37]	1	1	0	0	1	0	1	1	1	0	1	0	7
Ref. [34]	1	1	1	1	0	3	1	2	0	0	1	1	12
Ref. [38]	1	1	1	1	0	3	1	2	1	0	1	1	13
Ref. [40]	1	1	1	1	1	0	1	2	1	0	1	1	11
Ref. [32]	1	1	1	1	1	0	1	2	1	0	1	1	11
Ref. [25]	1	1	0	1	0	1	1	2	1	0	1	1	10
Ref. [24]	1	1	0	1	0	0	0	1	1	0	1	1	7
Ref. [27]	0	1	1	1	1	1	1	2	1	0	1	1	11
Ref. [28]	1	1	1	1	1	2	0	2	1	0	1	1	12
Ref. [39]	1	1	0	1	0	0	0	1	1	1	0	0	6
Ref. [31]	1	1	1	1	1	3	0	0	1	0	0	1	10
Ref. [22]	1	1	1	1	0	0	0	2	0	0	1	0	7
Ref. [41]	1	1	1	1	0	3	1	2	1	0	1	1	13
Ref. [29]	0	0	0	0	0	2	0	1	1	0	1	1	6
Ref. [23]	0	1	0	1	0	0	0	2	0	0	1	1	6
Ref. [33]	0	1	1	1	1	3	1	2	1	0	1	1	13
Ref. [8]	1	1	1	1	0	3	0	1	0	0	1	0	9
Ref. [21]	1	1	0	0	1	2	0	2	1	0	1	1	10
Ref. [35]	1	1	0	0	0	1	0	1	1	0	0	0	5
Ref. [30]	1	1	1	1	0	1	1	2	1	0	1	1	11

**Table 2 ijerph-19-15165-t002:** Demographics and clinical characteristics of the experimental and control groups of the trials included in the systematic review.

Study	Group	Subjects [N]	Sex [N Male; Female]	Age(y) [M; SD]	FEV_1_ L/%_pred_ [M; SD]	FVC_1_ L/%_pred_ [M; SD]
Ref. [36]	**TR**	100	35; 65	69.43; 7.38	-/39.55; 15.13	-
	**CI**	100	32; 68	68.87; 5.33	-/38.65; 13.68	-
Ref. [26]	**TR**	64	41; 23	69.10; 7.90	-/58.00; 23.60	-/88.40; 22.00
	**CI**	26	18; 8	71.40; 8.60	-/60.50; 20.10	-/83.20; 21.20
Ref. [37]	**TR**	27	16; 11	67.40; 10.20	-/36.10; 14.10	-/67.40; 19.90
	**CI**	27	15; 12	72.50; 7.40	-/32.80; 8.50	-/70.20; 17.00
Ref. [34]	**TR**	171	111; 60	66.00; 8.00	-/55.00; 20.00	-
	**NI**	172	108; 64	67.00; 8.00	-/57.00; 21.00	-
Ref. [38]	**TR**	46	30; 16	62.30; 8.20	-/42.39; 13.49	-/73.38; 15.88
	**CI**	48	33; 15	63.00; 6.60	-/42.93; 13.78	-/74.50; 15.47
Ref. [40]	**TR**	67	32; 35	68.40; 8.70	-/32.60; 10.30	-
	**CI**	67	28; 39	68.20; 9.40	-/33.70; 8.40	-
Ref. [32]	**TR**	80	48; 32	69.00; 13.00	-/52.00; 19.00	-/78.00; 17.00
	**CI**	86	51; 35	69.00; 10.00	-/49.00; 19.00	-/79.00; 22.00
Ref. [25]	**TR**	53	44; 9	70.92; 6.38	-	-
	**CI**	53	43; 10	71.83; 7.60	-	-
Ref. [24]	**TR**	12	10; 2	74.00; 8.00	-/58.00; 23.20	2.80; 0.70/-
	**CI**	15	14; 1	75.00; 9.00	-/60.60; 20.80	3.20; 0.70/-
Ref. [27]	**TR**	29	17; 12	68.00; 9.00	-/90.00; 8.00	-/104.00; 8.00
	**NI**	29	17; 12	67.00; 10.00	-/92.00; 7.00	-/104.00; 24.00
Ref. [28]	**TR**	10	4; 6	73.00; 4.00	1.00; 0.30/-	-
	**NI**	10	3; 7	67.00; 8.00	0.90; 0.20/-	-
Ref. [39]	**TR**	50	41; 9	65.90; 8.90	-/48.70; 11.20	0.00; 11.20/-
	**CI_1_**	50	38; 12	65.60; 8.80	-/50.40; 11.40	-/50.30; 11.40
	**CI_2_**	50	37; 13	64.80; 9.30	-/47.40; 11.50	-/48.40; 10.30
Ref. [31]	**TR**	29	21; 8	69.40; 3.30	-/49.20; 0.50	1.70; 0.00/-
	**NI**	28	23; 5	68.80; 1.40	-/49.80; 0.70	1.80; 0.00/-
Ref. [22]	**TR**	33	27; 6	66.40; 9.50	-/47.50; 23.30	-/68.70; 30.20
	**CI**	23	19; 4	71.30; 6.70	-/51.50; 23.90	-/79.10; 30.00
	**NI**	29	19; 10	70.80; 8.70	-/41.40; 18.40	-/69.90; 28.00
Ref. [41]	**TR**	22	19; 3	70.40; 9.40	-/61.00; 18.70	-
	**NI**	20	14; 6	65.06; 11.10	-/69.40; 24.00	-
Ref. [29]	**TR**	10	9; 1	69.20; 5.41	-/64.60; 11.97	-
	**NI**	10	10; 0	67.40; 6.60	-/60.40; 20.14	-
Ref. [23]	**TR**	17	7; 10	65.00; -	-/61.40	-
	**CI**	19	3; 16	64.00; -	-/58.00	-
Ref. [33]	**TR**	19	12; 7	73.00; 8.00	-/60.00; 23.00	-/89.00; 25.00
	**NI**	17	6; 11	75.00; 9.00	-/68.00; 19.00	-/98.00; 17.00
Ref. [8]	**TR**	47	44; 3	66.90; 9.60	-/49.60; 21.90	-/80.70; 20.20
	**CI**	50	38; 12	66.70; 7.30	-/51.80; 17.30	-/78.40; 18.40
	**NI**	50	37; 13	64.00; 8.00	-/51.70; 21.00	-/80.00; 20.30
Ref. [21]	**TR**	84	42; 42	62.00; 9.00	-/59.00; 20.00	-/101.00; 20.00
	**NI**	73	37; 36	63.00; 8.00	-/53.00; 15.00	-/99.00; 19.00
Ref. [35]	**TR**	55	21; 34	69.30; 7.80	0.40; 1.20/-	0.50; 0.20/-
	**NI**	65	36; 29	71.80; 8.10	0.30; 0.10/-	0.50; 0.20/-
Ref. [30]	**TR**	23	-	67.50; 6.20	-/46.70; 13.50	2.20; 0.70/-
	**CI_1_**	23	-	68.30; 7.00	-/42.60; 12.00	2.10; 0.80/-
	**CI_2_**	23	-	67.20; 7.30	-/47.20; 12.40	2.30; 0.60/-
	**CI_3_**	23	-	69.40; 6.40	-/48.20; 15.00	2.30; 0.80/-

Legend: CI, conventional intervention; COPD, chronic obstructive pulmonary disease; M, mean; N, number; NI, no intervention; SD, standard deviation; TR, telerehabilitation.

**Table 3 ijerph-19-15165-t003:** TR Descriptors, approach, and technology.

Study	FITT Descriptors	TR Approach				Technology
		Model *	Monitoring/Assessment	Decision	Feedback	
Ref. [36]	Frequency: 10–20 sessions/W × 12 W Intensity: - Time: 25–30 min Type: I, M (educational + Banduanjin exercise training)	A	Y	Y	Online	Platform: WeChat Device: mobile phone + PC Digital content
Ref. [26]	Frequency: 2–5 sessions/W × 6 W Intensity: increment of session length every W Time: - Type: I, M (educational + aerobic exercise training)	A	Y	Y	Online	Platform: My COPD Digital content
Ref. [37]	Frequency: 3–5 sessions/W × 8 W Intensity: - Time: - Type: I, U (aerobic/anaerobic exercise training)	A	Y	Y	Online	Platform: Virtual Autonomous Physiotherapist Agent platform Device: tablet/pc + pulsometer + biometric sensor Digital content
Ref. [34]	Frequency: 12 W Intensity: - Time: - Type: I, U (aerobic exercise training)	S	Y	Y	Offline	Platform: Fitbug Air coaching platform Device: step counter Digital content
Ref. [38]	Frequency: 3 sessions/W × 8 W Intensity: - Time: 120 min Type: I, M (educational + aerobic/breathing/weightlifting exercise training)	A	Y	Y	Online	Platform: TelePR platform Device: mobile phone + pulsometer Digital content
Ref. [40]	Frequency: 3 sessions/W × 10 W Intensity: Borg (score 4–7) + 40–80% of one repetition maximum (8–25 repetitions) Time: 35 min Type: G, M (educational + endurance exercise training)	S	Y	Y	Online	Device: videoconference system + touchscreen
Ref. [32]	Frequency: 2 sessions/W × 8 W Intensity: - Time: 30 min Type: I, M (Motivational + aerobic/strength/endurance exercise training)	A	Y	Y	Online	Device: pedometer + telephone
Ref. [25]	Frequency: 12 W Intensity: heart rate + conscious exertion score Time: 25–30 min Type: I, M (educational+ inspiratory muscles exercise training)	A	Y	Y	Online	Platform: Pulmonary Internet Explorer Rehabilitation platform + WeChat account Device: mobile phones + PC Digital content
Ref. [24]	Frequency: 7 sessions/W × 52 W Intensity: 30–40% maximal inspiratory muscle forces Time: - Type: I, M (educational+ aerobic/inspiratory muscles exercise training)	A	N	N	Offline	Device: pedometer fitted with actigraph
Ref. [27]	Frequency: 5 sessions/W × 8 W Intensity: Borg Time: 30 min Type: I, M (self-management + endurance exercise training)	A	Y	Y	Online	Device: pedometer + mobile phones
Ref. [28]	Frequency: 21 sessions/W × 8 W Intensity: 40–50% Pi max Time: - Type: I, U (inspiratory muscles exercise training)	A	Y	Y	No	Device: POWERbreathe KH2
Ref. [39]	Frequency: 2 sessions/W × 8 W Intensity: - Time: - Type: I, U (exercise training)	A	Y	Y	Offline	Platform: WeChat platform Device: mobile phones Digital content
Ref. [31]	Frequency: 16 W Intensity: - Time: - Type: I, U (inspiratory muscles exercise training)	A	Y	N	Offline	Platform: Dyspnea breathing program website Digital content
Ref. [22]	Frequency: 3 sessions/W × 12 W Intensity: 60–80% of maximal heart rate on 6 MWT Time: - Type: I, U (aerobic/strength exercise training)	A	Y	Y	Offline	Device: Heart monitor
Ref. [41]	Frequency: 1 session/W × 24 W Intensity: - Time: 60 min Type: G, M (educational + aerobic/stretching exercise training)	A	Y	Y	Online	Platform: self-management program Device: pedometer + smartphone Digital content
Ref. [29]	Frequency: 3 sessions/W × 8 W Intensity: Borg + no more than 10 repetitions for endurance exercises Time: 65 min Type: I, M (self-management + aerobic/endurance exercise training)	A	Y	Y	Online	Digital content
Ref. [23]	Frequency: 14 sessions/W × 12 W Intensity: Borg Time: - Type: I, U (inspiratory muscles exercise training)	A	Y	Y	Online	Device: mechanical threshold leadings breathing trainer + web-based SurveyXactn software
Ref. [33]	Frequency: 3 sessions/W × 8 W Intensity: Borg Time: 60 min Type: I, U (aerobic/strength exercise training)	S	Y	Y	Online	Device: PC with in-built camera + lower limb cycle ergometer + finger-tip pulse oximeter + real-time videoconferencing system
Ref. [8]	Frequency: 5 sessions/W × 52 W Intensity: - Time: 60 min Type: I, M (psychological support + self-management + breathing exercise training)	A	Y	Y	Online	Platform: TELECARE platform Device: multimodal apparatus wireless fitted with Bluetooth technology + tablet
Ref. [21]	Frequency: 7 sessions/W × 26 W Intensity: set by the website Time: - Type: I, U (aerobic exercise training)	A	Y	Y	Offline	Platform: website Device: embedded accelerometer in the smartphone
Ref. [35]	Frequency: - Intensity: - Time: - Type: I, M (educational + self-management + inspiratory muscles exercise training)	A	Y	N	Online	Platform: web platform Digital content
Ref. [30]	Frequency: 7 sessions/W × 8 W Intensity: set by the website Time: 50 min Type: I, U (aerobic exercise training)	A	Y	Y	Offline	Device: threshold inspiratory trainer + threshold expiratory trainer with monitoring device

Legend: d, day; FITT, intervention frequency intensity time and type; G, group sessions; I, individual sessions; W, week; N, No; Y, Yes; U, unimodal; M, multimodal; T, time; F, frequency; I, intensity. * The model of TR refers only to the physical component of rehabilitation.

### 3.4. TR Approach

*Model*: Nineteen studies presented an asynchronous [8,21,22,23,24,25,26,27,28,29,30,31,32,35,36,37,38,39,41] while three works presented a synchronous model [33,34,40]. 

*Assessment and Monitoring*: All studies except Kawagoshi et al. [24] included the monitoring of the rehabilitation at a distance using videoconference software [33,40], web-based platforms or applications [8,21,25,26,29,34,35,36,37,38,39,41], or devices, such as the pedometer (e.g., [27,32]), heart monitor [22], and breathing trainer device [23,28,30].

*Decision*: Nineteen trials provided decisions during the intervention to adapt the dose of the rehabilitation [8,21,22,23,25,26,27,28,29,30,32,33,34,36,37,38,39,40,41] while the rest of the interventions were not modified along the program period [24,31,35]. 

*Feedback*: Among the asynchronous interventions, 14 trials provided online feedback [8,23,25,26,27,29,32,33,35,36,37,38,40,41], 7 interventions used offline feedback [21,22,24,30,31,34,39], and 1 trial did not provide feedback during the program [28].

### 3.5. TR Technology

Among the 22 trials, 7 studies provided a comprehensive technological kit (healthcare platform, devices, and digital contents) [25,34,36,37,38,39,41]. In nine trials [22,23,24,27,28,30,32,33,40], the technology consisted exclusively of devices. Two trials adopted both platforms and devices [8,21], three studies used both platform and digital content [26,31,35], and one study [29] utilized only digital content.

### 3.6. TR Adherence and Safety

A small group of studies investigated adherence to TR and treatment safety. In total, 8/22 works registered the adherence rate to the treatment, reporting positive results. Among these, 5 studies highlighted an adherence rate of >80% in the TR group [8,23,24,37,38].

In total, 7/22 studies evaluated safety by reporting adverse events and hospitalization [26,34,37,38,39,40,41] that occurred during the intervention. In six works, the adverse events were absent (0%) [37,38] or minimal (1–3%) [26,39,40,41] while one study reported that 27% of patients experienced safety issues [34].

### 3.7. Efficacy at the End of Rehabilitation

Outcome measures evaluated in each trial are described in Table 4.

*Exercise capacity*: The overall effect of TR compared to NI, computed from seven studies, was significant and small (*g* = 0.29, 95% CI = 0.08, 0.49, *p =* 0.001; Figure 2). The heterogeneity across studies was moderate (I^2^ = 52.00, Q = 3.78, *p* = 0.71) and the funnel plot showed three estimated missing studies on the left side (see Appendix A). The asynchronous TR showed a moderate and significant effect on NI (5 studies included; *g* = 0.39, 95% CI = 0.10,0.68, *p =* 0.001; Figure 2), with null true heterogeneity (I^2^ = 00.00, Q = 1.16, *p* = 0.88) and one estimated missing study (on the right side of the funnel plot, see Appendix A), while the synchronous TR had a moderate but insignificant effect on NI (2 studies included; *g* = 0.31, 95% CI = −0.27, 0.89, *p >* 0.05; Figure 2).

Concerning CI, TR showed no inferiority effect (4 studies included, *g* = 0.12, 95% CI = −0.23, 0.46, *p >* 0.05; Figure 3), with a null true heterogeneity (I^2^ = 00.00, Q = 0.92, *p* = 0.82), and one missing study estimated (on the left of the funnel plot, see Appendix A). All the studies included in the analysis adopted an asynchronous approach.

### 3.8. Dyspnoea

The overall effect of TR compared to NI, computed from four studies, was significant and large (*g* = 0.76, 95% CI = 0.15, 1.37, *p =* 0.01; Figure 4). The true heterogeneity across studies was moderate (I^2^ = 67.11, Q = 9.08, *p*= 0.03), and the funnel plot showed two estimated missing studies on the right side (see Appendix A). Considering the three studies with an asynchronous approach, TR showed a large and significant effect on NI (*g* = 0.82, 95% CI = 0.03, 1.61, *p =* 0.01; Figure 4), with high true heterogeneity (I^2^ = 77.59, Q = 8.19, *p*= 0.02), and two estimated missing studies (on the right side of the funnel plot, see Appendix A). The only study that adopted synchronous TR had a medium effect on NI (*g* = 0.48, 95% CI = −0.43, 1.39; Figure 4).

Considering the seven studies that tested the effect of TR on CI, TR revealed a significant small effect on AC (*g* = 0.27, 95% CI = 0.08, 0.47; *p* = 0.001; Figure 5). The true heterogeneity was null (I^2^ = 0.00, Q = 1.90, *p* = 0.93), and two studies were estimated to be missing (on the right side of the funnel plot, see Appendix A).

### 3.9. Quality of Life

The overall effect of TR compared to NI, computed from four studies, was significant and medium (*g* = 0.57, 95% CI = 0.15, 1.00, *p =* 0.001; Figure 6). The true heterogeneity across studies was moderate (I^2^ = 68.21, Q = 20.67, *p* < 0.01), and the funnel plot suggested one estimated missing study on the left side (see Appendix A). Considering the five studies with an asynchronous approach, TR showed a large and significant effect on NI (*g* = 0.60, 95% CI = 0.13, 1.07, *p =* 0.01; Figure 6), with moderate true heterogeneity (I^2^ = 54.74, Q = 8.78, *p* = 0.07) and no estimated missing studies (see Appendix A). The two studies adopting synchronous TR showed a non-inferiority effect on NI (*g* = 0.15, 95% CI = −0.25, 0.55; Figure 6) and very low heterogeneity (I^2^ = 17.08, Q = 1.21, *p* = 0.27).

Considering the six studies that tested the effect of TR on CI, TR showed a non-superior effect on CI (*g* = 0.05, 95% CI = −0.28, 0.39; Figure 7). The true heterogeneity was low (I^2^ = 42.96, Q = 8.48, *p* = 0.13), and three studies were estimated to be missing (on the right side of the funnel plot, see Appendix A). All the studies included in this analysis adopted an asynchronous approach of TR.

## 4. Discussion

This review aimed to outline the TR approach adopted for people with COPD in RCTs regarding the intervention descriptors, actions, and type of technology. Moreover, the effects of TR on medical-benefit outcomes compared to no intervention (NI) and the conventional intervention (CI) were tested.

Overall, 22 studies were included in the systematic review, presenting a sample of 2234 subjects with a mild-to-moderate COPD condition (GOLD I–III). To date, people in the earliest phases of COPD encounter difficulties in benefiting from treatment in the clinic, which usually targets a small fraction of people at a severe stage of illness. In this view, TR opens the potential to go beyond the accessibility constraints and adherence issues, such as structural barriers, related to the clinical setting [10,42] and may move the COPD rehabilitation target from the severe to the earliest phases toward an early intervention perspective, assuring patient’s compliance and affordability.

Concerning the treatment approach, about 60% of the included studies provided a multidimensional intervention, including educational and self-management components in addition to exercise training, while the rest of the studies were unidimensional and focused only on exercise training. In line with the pulmonary rehabilitation statements supporting the need for comprehensive treatment for COPD, 3–4 times/week [6], this result was partly below expectations and probably linked to the complexity and maturity level of the available technological facility. Future TR models should propose an integrative and multidisciplinary approach in line with the recent multidimensional rehabilitation [43], favoring comprehensive care through the utilization of digital healthcare platforms, and may focus on the ideal dose of the TR program. To date, the exercise dose largely varies in TR for COPD, in terms of the duration of treatment (from 6 to 52 weeks in total [8,24,26], frequency (from seven sessions per day to once a week [30,41]), time per session (from 10 to 120 min [37,38]), and intensity, which is determined and adapted according to different modalities [23,24,25,26,34].

Concerning the TR approach, almost all the studies provided monitoring, with a prevalence of 86% of the trials adopting the asynchronous model and 41% of the trials adopting online feedback. The asynchronous modality assures considerable advantages, such as overcoming the 1:1 patient therapy setting and moving the rehabilitation service to a broader target population [44]. Interestingly, recent trials published in 2021 and 2022 presented a complex and comprehensive technological system and an asynchronous model, suggesting that technological maturity is becoming a TR prerequisite. The asynchronous model may reduce the healthcare overburden and related costs, which are expected to double by 2030 due to the increase in COPD prevalence [45]. Notably, the flexibility of asynchronous TR may positively impact compliance and adherence to the treatment, a critical issue for COPD. Especially, although only a small group of studies have focused on adherence to treatment, good adherence to TR has been reported. Moreover, seven studies investigated the safety issue related to TR, reporting an absence of or minimal adverse events in 6 studies, with only one work reporting 27% adverse events, 5 of which led to hospitalization. Considering the crucial role of safety in rehabilitating fragile patients, such as COPD, and in light of a translational perspective, these data are promising. Nevertheless, the majority of studies in the review did not describe adverse events and the lack of reporting in these works does not imply that no adverse events occurred. More contributions are needed with a specific focus on safety constraints related to TR [14].

Our meta-analyses showed a significant increase in functional capacity, dyspnea, and quality of life for people with mild-to-moderate COPD after TR compared to no rehabilitation. Furthermore, concerning conventional intervention, TR was revealed to be equally effective on functional capacity and quality of life, and more effective than CI on dyspnea, the most disabling symptom related to COPD. Additionally, TR positively impacted COPD outcomes, showing the transfer of positive benefits into everyday life. These findings support TR as a potentially valid integration into clinic rehabilitation when distance barriers and accessibility constraints occur. Indeed, it is equally effective on quality of life, with additional advantages, such as lowering the user’s costs related to transport, spaces, and in-person services for rehabilitation. Therefore, future directions can tailor the modality of the continuity of care to the patient’s needs, choosing between in-clinic rehabilitation and TR as equally valid solutions.

Importantly, our meta-analysis results revealed that the asynchronous TR interventions had a higher effect on exercise capacity, dyspnea, and quality of life than synchronous treatments. This finding may be related to the advantage of this intervention modality, providing a more intense and long-term treatment that does not require the online (*face-to-face*) presence of the therapist but guarantees the patient’s rehabilitative activities by monitoring in an offline mode. For instance, Vasilopoulou et al. [8] and Santiworakul et al. [29] provided an asynchronous intensive treatment consisting of 3–5 sessions/week, in line with the statements for pulmonary rehabilitation [6]. Overall, the duration of the benefit of TR remains an overlooked issue that needs further investigation.

Some essential caveats of this study need to be mentioned: the trials selected for this review were largely heterogeneous in terms of the intervention characteristics and TR approach and technology, potentially affecting the meta-analysis findings. Finally, the TESTEX scale results revealed the low quality of four studies selected for this review, reporting critical issues such as the absence of the assessor’s blindness and the intention-to-treat method.

## 5. Conclusions

In conclusion, the TR approach for COPD currently shows a heterogeneous scenario. The effect of TR, especially with asynchronous monitoring, increases and maintains functional capacity, breath, and quality of life, and favors adherence in people with COPD. However, currently, there is a lack of consensus on the essential elements and features, which this approach should own to guarantee comprehensive and adaptable continuity of care. Future works need to clarify the fundamental role of feedback, monitoring, and decision-making in the TR model and afford a common vocabulary when referring to home-based or TR treatment.

## Figures and Tables

**Figure 1 ijerph-19-15165-f001:**
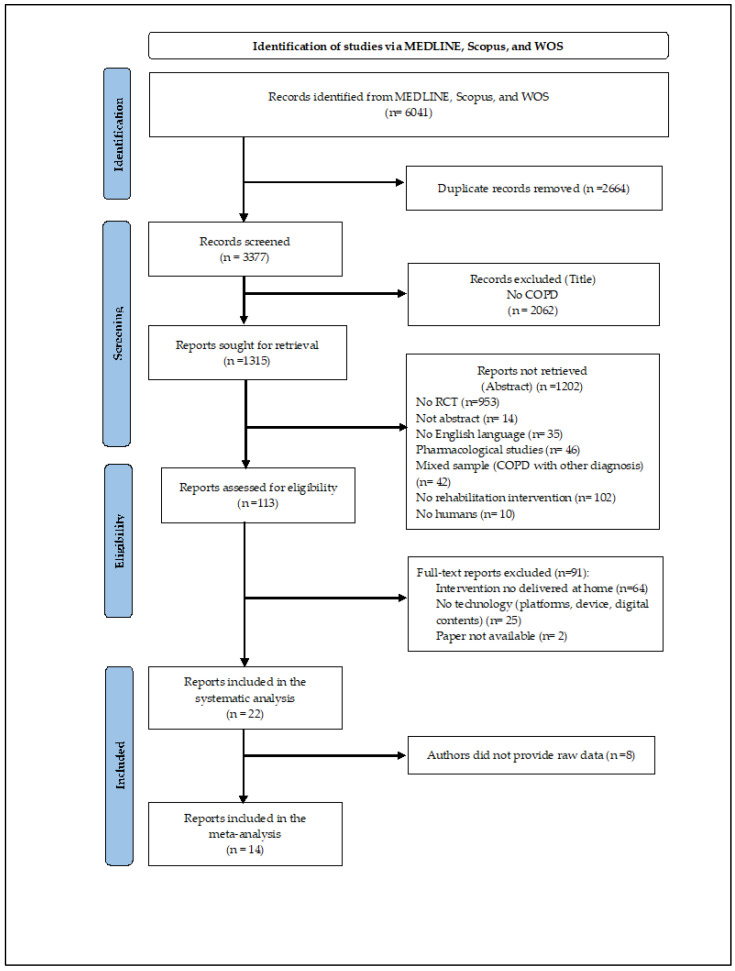
2020 PRISMA Flow Diagram.

**Figure 2 ijerph-19-15165-f002:**
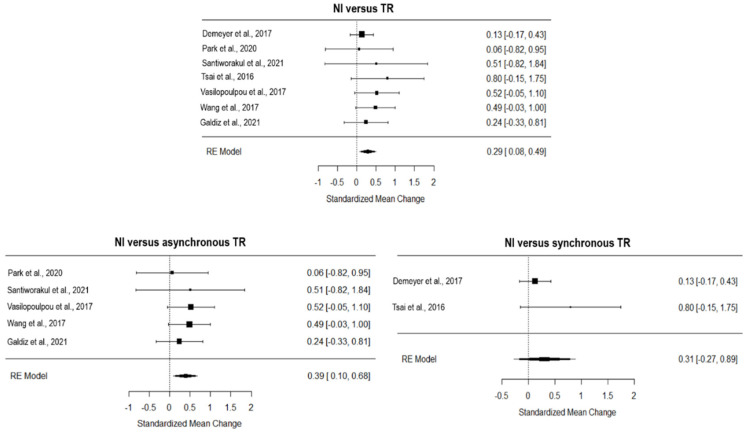
Forest plot on the effect of TR on functional capacity compared to NI. TR, TR; NI, no intervention [8,29,33,34,35,38,41]. Ref. [31] was not included in the analysis based on the leave-one-out analysis.

**Figure 3 ijerph-19-15165-f003:**
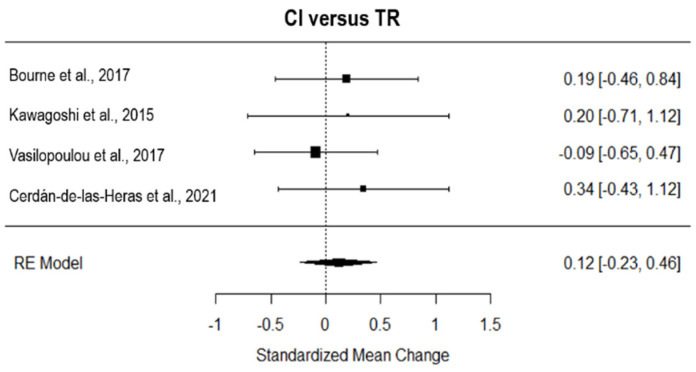
Forest plot of the effect of TR on functional capacity compared to conventional intervention. CI, conventional intervention; TR, TR [8,24,26,37].

**Figure 4 ijerph-19-15165-f004:**
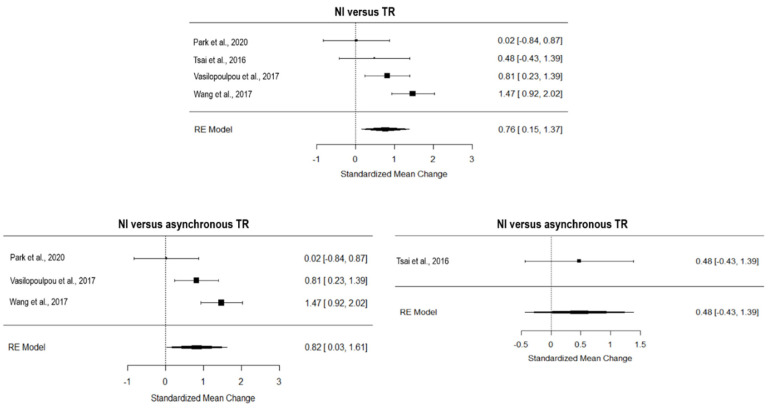
Forest plot of the effect of TR on dyspnea compared to usual care. TR, TR; NI, no intervention [8,33,35,41]. Ref. [31] has not been included in the analysis based on the leave-one-out analysis.

**Figure 5 ijerph-19-15165-f005:**
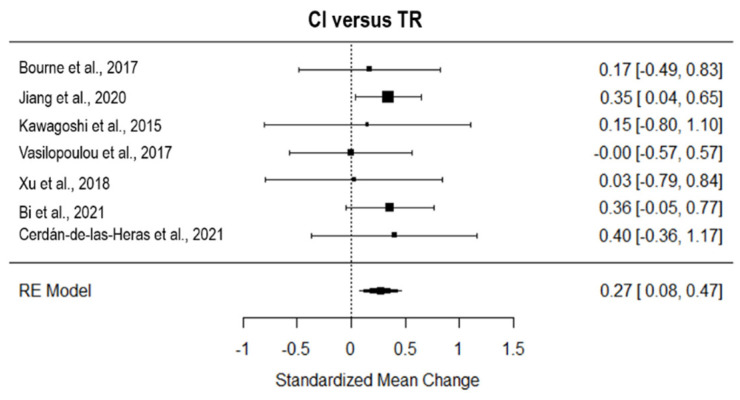
Forest plot of the effect of TR on dyspnea compared to conventional intervention. CI, conventional intervention; TR, TR [8,24,25,26,30,36,37].

**Figure 6 ijerph-19-15165-f006:**
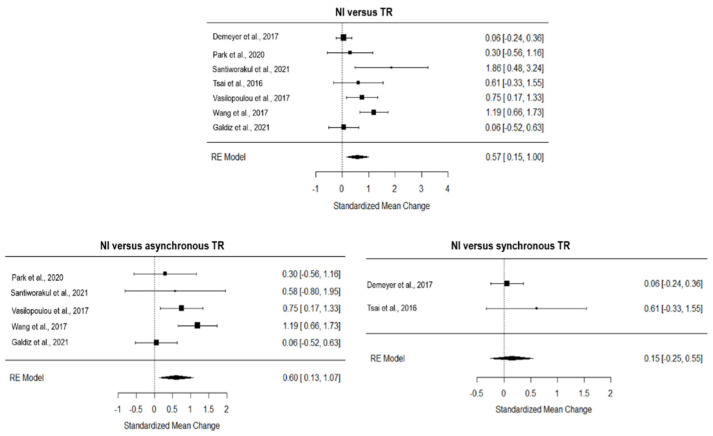
Forest plot of the effect of TR on quality of life compared to usual care. TR, TR; NI, no intervention [8,29,33,34,35,38,41]. Ref [31] was not included in this analysis based on the leave-one-out analysis.

**Figure 7 ijerph-19-15165-f007:**
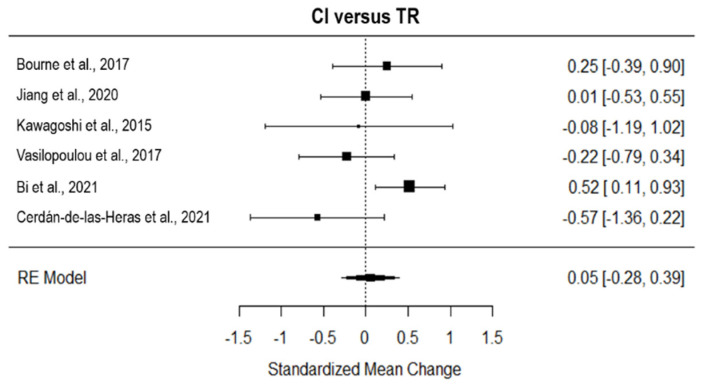
Forest plot of the effect of TR on quality of life compared to conventional intervention. CI, conventional intervention; TR, TR [8,24,25,26,36,37].

**Table 4 ijerph-19-15165-t004:** Outcome measures of studies included in the systematic review.

Outcome	Domain	Subdomain	Tool	Pre-Based	Prom-Based	References
Medical Benefit	Functional Capacity	Endurance	6 min Walk Distance	x		[8,21,22,23,24,26,27,29,30,31,32,33,34,35,37,38,39,40,41]
			Endurance shuttle walk test	x		[33]
		Physical Activity	Actigraph	x		[8,21,24,27,33,34,37,40,41]
			Physical Activity Level		x	[40]
			Incremental Shuttle Walking Test	x		[33]
			Functional Performance Inventory—Short Form		x	[33]
		BMI	Weight × height^2^	x		[21,22,30]
		Strength	Quadricep force	x		[29,34]
			Triceps force	x		[29]
		Breath	Modified Medical Research Council dyspnea scale		x	[8,22,24,25,27,28,30,32,33,34,35,39]
			San Diego Shortness of Breath Questionnaire		x	[41]
			Spirometry values (i.e., FEV_1_, FVC)	x		[21,22,23,24,28,30,31,33,35,36,37,39]
	Participation	Quality of Life	St George’s Hospital Respiratory Questionnaire		x	[8,25,30,31,35,37]
			COPD-CAT		x	[8,24,26,30,33,34,36,39,40]
			Chronic Respiratory Questionnaire		x	[21,24,27,32,33,38]
			EuroQol 5-Dimension Questionnaire		x	[40]
			36-Item Short-Form Health Survey		x	[38,41]
			Clinical COPD Questionnaire		x	[29,40]
			IADL		x	[37,39]
		Mood	Hospital Anxiety and Depression Scale		x	[26,30,32,33,40]
			Profile of Mood States-Short Form		x	[41]
			General Anxiety Disorder Score		x	[37]
			Beck Depression Inventory		x	[39]
			State-trait Anxiety Inventory		x	[39]
		Self-Efficacy	Exercise Self-Regulatory Efficacy Scale		x	[25]
			Self-Efficacy for Managing Chronic Disease 6-Item Scale		x	[41]
			Alberto Chronic Obstructive Pulmonary Disease Self-Care Behavior Inventory		x	[41]
			Pulmonary Rehabilitation Adapted Index of Self-Efficacy		x	[32,33]
		Social Support	MOS Social Support Survey		x	[41]
	Mortality	-	N Deaths			[40]
	Morbidity	-	COPD-specific COmorbidity TEst		x	[32]
			Exacerbation history and comorbidity		x	[34]
Patient-Relevant	Adherence	-	N sessions/expected N sessions × 100	x		[8]
			N sessions performed	x		[23,26,33,37,41]
			Time dedicated to tasks	x		[41]
			Work rate/exercise time (s)	x		[33]
			N participants who complied with treatment (at least 8 consecutive weeks)	x		[38]
	Safety	-	Adverse events and hospitalizations		x	[26,34,35,37,38,39,41]
			Frequency of ED visits, outpatient clinic visits		x	[39,41]
			Open-ended questions		x	[41]

## Data Availability

The protocol was registered in the International Prospective Register of Systematic Reviews (PROSPERO): registration ID CRD42021277381.

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
