# Peer review of "Characteristics, Components, and Efficacy of Telerehabilitation Approaches for People with Chronic Obstructive Pulmonary Disease: A Systematic Review and Meta-Analysis"

_ijerph, 2022, doi:10.3390/ijerph192215165_

Round 1
Reviewer 1 Report
Comment 1. The algorithm is unclear. Please rewrite
Comment 2. Two studies which including of your manuscript have bias of methods
Comment 3. "The search was performed on 26th September 2022, including all studies published from 2010 to September 2022" It is very interest you can reading so many studies and writing the article in approximately 20 days.
Comment 4. "Eligibility criteria.....people with COPD (⩾18 years old)" Why selected this age ?
Comment 5. It would be interesting to report the effect of medication
Author Response
Comment 1. The algorithm is unclear. Please rewrite
Answer to comment 1: Whether for the algorithm is meant the search string, we modified it by reporting the keywords utilized. : Whether for the algorithm is meant the criteria of screening and eligibility we added more details in the figure 1 and the text.
Comment 2. Two studies which including of your manuscript have bias of methods
Answer to comment 2: We thank the reviewer. We checked table 1 and the information in the text. Now the Table and the text report coherent data.
Comment 3. "The search was performed on 26th September 2022, including all studies published from 2010 to September 2022" It is very interest you can reading so many studies and writing the article in approximately 20 days.
Answer to comment 3: This review has been registered in PROSPERO on 26th October 2021 (ID CRD42021277381). We decided to update the search on 26th September, integrating the new works published.
Comment 4. "Eligibility criteria.....people with COPD (⩾18 years old)" Why selected this age ?
Answer to comment 4: We reported this specific eligibility criterion to include in the review only studies on the adult population.
Comment 5. It would be interesting to report the effect of medication
Answer to comment 5: In all the studies, the telerehabilitation intervention has been provided as an add-on to the pharmacological treatment. The results showed in the meta-analysis refer to the effect of the non-pharmacological intervention in addition to the stable medication treatment.
Reviewer 2 Report
In my view, this article is very well concieved: the summary, is very clear with a description of the main results; the introduction refers to the current state of the art regarding the focus under study, telerehabilitation for patients with COPD; the methods are described in a reproducible way, presenting a PRISMA flowchart for sample identification, both for meta-analysis (n14) and for systematic review (n22); the results were presented in detail by presenting the respective tables.
However, some of the tables could be less exhaustive and contain only the necessary information and in a more synthetic way, for example, table 2, the columns of the exclusion and inclusion criteria could be removed, as these have been explained earlier. Table 3 could be removed (too long and unappealing visualization) since the description of these results are presented in a clear and objective way, or to place the table structured differently, for example, not article by article, but only the summary of the results of the types of interventions in TR.
The discussion is well presented, identifying the limits of this study, such as the fact that in the sample they identified a diversity of telerehabilitation formats, which reflects the need to find consensus to better study the effects of this on functional capacity, breathing and quality of life in COPD patients. Therefore, this study leaves this challenge in terms of future research scenarios in this area.
Author Response
We thank the reviewer for the suggestions.
We now removed the inclusion and exclusion criteria in Table 2 and move them to a new table in Supplementary Material (Table S1).
We synthetized and structured differently Table 3. We believe that this table is highly informative for the purpose of the review, which is the profiling of the telerehabilitation model adopted for COPD (see Table 3).